# Amylin and Secretases in the Pathology and Treatment of Alzheimer’s Disease

**DOI:** 10.3390/biom12070996

**Published:** 2022-07-17

**Authors:** Som Singh, Felix Yang, Andy Sivils, Victoria Cegielski, Xiang-Ping Chu

**Affiliations:** Department of Biomedical Sciences, School of Medicine, University of Missouri, Kansas City, MO 64108, USA; somsingh@mail.umkc.edu (S.S.); felixyang@mail.umkc.edu (F.Y.); andysivils@mail.umkc.edu (A.S.); vicbmy@umkc.edu (V.C.)

**Keywords:** amylin, biomolecules, Alzheimer’s disease, secretases, modulation

## Abstract

Alzheimer’s disease remains a prevailing neurodegenerative condition which has an array physical, emotional, and financial consequences to patients and society. In the past decade, there has been a greater degree of investigation on therapeutic small peptides. This group of biomolecules have a profile of fundamentally sound characteristics which make them an intriguing area for drug development. Among these biomolecules, there are four modulatory mechanisms of interest in this review: alpha-, beta-, gamma-secretases, and amylin. These protease-based biomolecules all have a contributory role in the amyloid cascade hypothesis. Moreover, the involvement of various biochemical pathways intertwines these peptides to have shared regulators (i.e., retinoids). Further clinical and translational investigation must occur to gain a greater understanding of its potential application in patient care. The aim of this narrative review is to evaluate the contemporary literature on these protease biomolecule modulators and determine its utility in the treatment of Alzheimer’s disease.

## 1. Introduction

Across the globe, Alzheimer’s disease (AD) is a prevailing cause of dementia and death in elderly populations [1,2]. Clinically, it presents in three stages of cognition sequencing from normal ability to functional impairment and memory loss, colloquially known as dementia [3]. However, recognizable pathologies of AD can be seen up to twenty years prior to onset of clinical symptoms [4]. One of the earliest pathologic hallmarks is accumulation of extracellular amyloid β (Aβ) in the cerebrum [5]. Additionally, neurofibrillary tangles composed of hyperphosphorylated tau are another core histopathologic feature of AD [6,7]. These protein deposits have thus served as major research concentrations for understanding the pathophysiology of this disease. The current school of thought centers around the amyloid cascade hypothesis, which proposes that Aβ deposition incites a succession of culminating protein debris and subsequent neuronal dysfunction [8]. Consequently, modern treatments aimed at modifying disease progression target different aspects of such protein aggregation and modulation [9].

### 1.1. Amyloid Cascade Hypothesis

The amyloid cascade hypothesis is the reigning theory which attempts to mechanistically explain the onset and pathophysiology of AD [10]. It was initially proposed in the late 20th century following the discovery that mutations on chromosome 21 involving dysfunctional amyloid precursor protein (APP) metabolism led to toxic Aβ peptide deposition [11]. In more detail, APP is a single protein with cleavage sites by secretase enzymes [12]. In AD, some of the cleaved protein fragments assemble into amyloid and that amyloid accumulates [12]. These accumulated protein aggregates may then become toxic, and are known to modify synaptic function, brain development, inhibit long-term potentiation (LTP), induce neurodegeneration, and prompt gliosis [13,14,15]. The beta-amyloid product received the initial attention of the amyloid hypothesis (AH) due to a pilot study by Schenk et al. in 1999 which found that immunization of transgenic mice with Aβ42 prevented beta-amyloid plaque formation, neural dystrophy, and astrogliosis [16]. However, multiple trials targeting these beta-amyloid proteins have been conducted since the work by Schenk et al., and no treatment directed at lowering Aβ plaques has been successful [17]. Other works in the literature claim that investigations into the AH over the past two decades have not accurately assessed the hypothesis [18]. They ascertain that those clinical trials have targeted Aβ monomers or plaques not associated with neurotoxicity. Instead, trials should have more precisely targeted Aβ oligomers [18]. Even the antibodies generated as purported treatments are argued to be inefficient tests of the treatment mechanism [18]. Nevertheless, the field in total appears to be nearing the end of the road with the AH, as it has been known since the late 20th century. Scientists and medical providers are now working to usher in a new era that may build off the idea that amyloid cascade products are essential in AD pathophysiology but will have to take a new road in doing so. One of these roads may be in the development of protein biomolecules and synthetic therapeutics.

### 1.2. Small Peptide Synthetic Therapy in Neurodegenerative Disease

The use of small peptide has a well-established history of therapeutic applications for the treatment of various neurologic pathology [19,20]. Applications of small peptide therapy include potential treatment for specific neurodegenerative diseases including AD, Parkinson’s disease, and Huntington’s disease [21]. With regard to AD, this condition is the most common fatal neurodegenerative disease across the globe, and several small peptides have already been investigated to have inhibitory effect on both Aβ and disease-relevant tau protein. One such peptide is KLVFF which has been shown to have inhibitory properties on Aβ aggregation, and has been strategically manipulated in order to disrupt pathogenic peptide aggregation [22]. More recently, Aillaud et al. found the novel D-amino acid peptide, ISAD1, successfully inhibited the fibrillization of tau protein [23]. In addition, studies have also demonstrated a high level of expression of ion channels in microglia implicated in AD that are potentially targetable by synthetic peptide therapy [24]. One model demonstrated that the voltage-gated potassium channel, Kv1.3, was highly expressed in human AD brains and sensitive to its potent and selective blocker 5-[4-Phenoxybutoxy] psoralen (PAP-1) small peptide therapy [25]. Moreover, oral PAP-1 proved to collectively reduce AD neurological deficits via reduction of neuroinflammation, cerebral amyloid load, and amyloid-associated pathologies [25]. However, among these various peptide-based investigations, scientists have hypothesized manipulation of enzymes which impact APP proteins implicated in the amyloid cascade hypothesis could be useful in the treatment of AD. Among the relevant enzymes are three proteases: alpha (α), beta (β), and gamma (γ)-secretases [26]. These three enzymes represent the three different locations where APP can be cleaved: the C-terminal of the Aβ domain via γ-secretase, the N-terminal of the Aβ domain via β-secretase, and within the Aβ domain via α-secretase [27].

## 2. Alpha Secretase Activators

Alpha secretase completes proteolysis of APP which leads down the non-amyloidogenic pathway, where beta secretase completes proteolysis of APP through the amyloidogenic pathway (Figure 1) [28]. Evidence suggests that α-secretase activity is modulated by metalloprotease inhibitors and metal ions, in fact three members of the ADAM (a disintegrin and metalloprotease) family are reported to be candidate α-secretases [28]. ADAMs are type I transmembrane proteins which will be proteolytically active if they have the relevant catalytically active domain [29].

Specifically, investigations have revealed that ADAM10 is the α-secretase that mediates the non-amyloidogenic pathway primarily, while APP cleavage by the β-secretase BACE1 or γ-secretase complex yields the pathogenic Aβ peptide [29]. However, another α-secretase is implicated in the pathology of AD, ADAM17 [30]. A genomic investigation showed that there was a single rare nonsynonymous variant in ADAM17 that co-segregated with an autosomal-dominant pattern of late onset AD in one family [30]. Furthermore, data suggested a strong negative correlation between APP gene expression and ADAM17 in the human brain [30]. Which is further captured by the fact that a p.R215l mutation of ADAM17 led to elevated Aβ formation in vitro [30].

Some investigations born from the idea that α-secretase activation could be therapeutic in AD treatment have been fruitful. In fact, one modifier of α-secretase via the retinoic acid pathway is the vitamin A derivative retinoic acid [31]. The idea here is that retinoic acid activates ADAM10 and thus reduces the amount of Aβ peptide [31]. Evidence shows that vitamin A deficiency leads to an increase in Aβ peptide levels in wild-type (WT) mice [31]. Even more interesting, the rescue of this deficiency has led to increased non-amyloidogenic processing [31].

As an extension of this idea, some researchers looked at the peroxisome proliferator-activated receptor (PPAR-α) which is another component of the retinoic acid pathway. PPAR-α has been demonstrated to activate ADAM10 transcription, eventually leading to a reduced production of pathogenic Aβ peptides [31]. However, there have been three clinical trials related to vitamin A supplementation in AD and only one of them had positive findings—which included only a maintained baseline cognitive performance over 12 months without any improvement [32].

However, there is some promise in utilizing the lipid-lowering medication gemfibrozil which activates PPAR-α and was shown to both inhibit the production of Aβ via upregulation of ADAM10 and stimulate cellular clearance by inducing lysosomal biogenesis in the 5XFAD transgenic model of AD [33,34,35,36,37]. In a phase II parallel-design, double-blind, placebo-controlled trial targeting predementia AD, there was a significant decline on the CANS-MCI cognitive battery, and almost statistically significant declines in Aβ 42 levels and hippocampal atrophy [38]. These findings tentatively validate the pathophysiologic hypothesis underpinning α-secretases as treatment or prevention for AD, but it is fair to say the evidence is less than convincing that these may be formidable treatments.

Other drugs already approved for treatment of AD have been found to increase α-secretase activity, such as selegiline the monoamine oxidase inhibitor which slows the progression of AD [39]. In fact, the antihypertensive therapy, atorvastatin was found to be somewhat protective as an AD treatment and it was discovered that it appears to induce activation of α-secretase [40]. Interestingly, evidence shows that ADAM10 levels are reduced in the cerebrospinal fluid (CSF) of AD patients, suggesting a natural inhibitor of the enzyme might be present in the pathology [41]. Two potential culprits are the secreted-frizzled-related protein 1 (SFRP1) and 2 (SFRP2) identified by Esteve et al. in 2011 [42,43]. Further research by Esteve et al. demonstrated in 2019 that SFRP1 is significantly increased in the brain and CSF of patients with AD [44]. Expanded investigation showed that decreasing SFRP1 function thus lowers AP accumulation, improves AD-related histopathological traits, and prevents LTP loss and cognitive deficits [44]. Thus, while the evidence is fairly new, SFRP1 has emerged as a potential therapeutic target for AD precisely because of its impact on α-secretase. This has ushered in a new focus on α-secretase disinhibitors, rather than just stimulators of the protease.

## 3. Beta-Secretase Inhibitors

Another transmembrane aspartic protease is β-secretase (BACE1 and BACE2), which serves as another enzyme that is implicated in the aggregation of Aβ plaque [45]. In support of the Aβ amyloid hypothesis for AD pathogenesis, BACE1 is theorized to cleave APP into Aβ plaques within Golgi endosomes and lysosomes, thus making the enzyme a potential modulation site to reduce neurodegenerative effects of Aβ plaques [46]. BACE1 has a crystalline structure with different structural conformations dependent on its activation/inactivation [45]. More specifically, BACE1 activity is highly associated with the flexibility of the flap covering the active binding site of the protease [47]. The importance of BACE1 has been seen in several gene knock-out (KO) studies done by researchers in attempts to observe any reduction of Aβ plaque with gene deletions. One such study demonstrated that partial BACE1 gene deletion (50% to 70%) induces synaptic plasticity deficit in adult mice [48]. Similarly, mouse-models with BACE1 gene KO alterations reported less Aβ build up in cerebral tissue [49]. While these evidence supports the idea that inhibition of the β-secretase protein has positive downstream effects in reducing Aβ plaque, complete lack of the BACE1 has also been shown to have putative detrimental effects to normal neurophysiology [50]. Although BACE1 gene was first identified through its critical role in AD pathogenesis, it is also a vital contributor of muscle spindle fiber formation and maturation [50]. Without the protein expressed in KO mice, researchers found unnatural alteration in mice movement due to impaired muscle proprioception and coordination [50].

The majority of β-secretase inhibitor development and testing surrounds the inhibition of BACE1. Its close homologue, BACE2, is not highly expressed in brain tissue and is instead found at higher concentration in pancreatic islet cells and a variety of other peripheral tissues [51]. Due to this, selective BACE2 inhibitors developed by researchers primarily concern the treatment of type 2 diabetes mellitus (T2DM) and but not AD [52]. However, several strides have been made to establish BACE2′s role as a potential contributor to AD risk despite not having significant documentation of its function within the central nervous system (CNS). Recent findings identified BACE2 to have conditional β-secretase activity that is dependent on a mutation in the juxtamembrane helix domain of the protein [53]. These results suggest a potential AD therapeutic that targets BACE2 gene without the adverse effects observed in inhibition of BACE1 gene [54]. Considering the expression sites of BACE2 gene, data suggests that it is possible for dual pharmacological targeting of the β-secretase for both T2DM and AD [53]. BACE2 gene may also alternatively act as a Aβ protease in addition to its better-understood function as an APP protease [55]. Overexpression of BACE2 gene may also have anti-amyloidogenic effects as suggested by Sun et al. [56]. It is important to note that cross-reactivity between BACE1 and BACE2 exists, and thus, therapeutics targeting BACE1 may also have unpredicted effects in BACE2 pleiotropism for better or for worse [55,56,57,58].

After over 2 decades of pharmacological research surrounding APP secretases, β-secretase is believed to be a modulation site that has immense potential in future drug development. The primary mode of inhibition of BACE1 gene is competitive, reversible inhibition through non-covalent interactions of small molecular drugs [59]. Similar to other aspartic proteases implicated in Aβ production, BACE1 gene contains several aspartic acid residues within its active binding site [59]. One such category of small molecular inhibitors is pseudopeptide β-secretase inhibitors [60]. OM99-1 and OM99-2 were the very first inhibitors of this class and were proven to be highly potent inhibitors of BACE1 [61]. After several years of adjusting various moieties of pseudopeptide β-secretase inhibitors, other groups introduced KMI-570 and KMI-684 which, at concentrations of 100 μM/L, displayed 84% β-secretase inhibition in cultured cells [62]. While initial results from these tests show promise, the pharmaceutical viability for these drugs are low as they have low oral availability and fail to cross the blood-brain-barrier (BBB) [63].

Verubecestat is another β-secretase inhibitor, which primarily elicits its inhibitory mechanism via high affinity binding of hydrophobic subsites of BACE1 gene with minimal binding to other human receptors, ion channels, or enzymes [64,65]. On a study done with mouse and primate models, in vitro effects of single oral doses of verubecestat were observed to have significant reduction in CSF and cerebral Aβ protein [64]. Further time course studies in primate models measured peak reductions of Aβ40 and sAPPB by 12–24 h that were sustained for 24 further hours [64]. While these results suggest promising applications on AD patients with large deposits of Aβ plaque, phase III clinical trials of mild to moderate AD revealed systemic adverse effects [66]. Referencing to previous studies regarding the physiological functions of BACE1, it is possible that verubecestat’s high affinity inhibition of BACE1 may have led to detrimental impairment of neurological and musculoskeletal processes [50,66]. One small-molecule active-site inhibitor of BACE1, LY288671, demonstrated potent inhibition of the target protease with minimal pleiotropism towards cathepsin D [67]. Although phase II clinical trials for this drug were discontinued due to concerns of hepatotoxicity, data gathered from in vitro human testing collectively highlight the potency of this drug while having no significant adverse effects elsewhere in the body [67].

Despite the discontinuation of phase II trials for LY-288671, a recent review of the drug’s therapeutic effects in rodent neuronal cultures show some promise in reducing Aβ secretion without impairment of synaptic transmission [68]. In a 10 day, in vitro test of three β-secretase inhibitors (LY-2886721, BACE inhibitor IV, and lanabecestat), these β-secretase inhibitors only decreased synaptic transmission at drug dosages high enough to inhibit Aβ30 and Aβ42 secretions by more than 50% [68]. More specifically, LY-288671 only impacted synaptic transmission when given at 3 μM, a concentration that decreased Aβ42 by 50% and Aβ40 by 50% or more [68]. It is possible that adjustments to the dosage and duration for this drug can be made for future pharmaceutical application since the balance between therapeutic and adverse effects is crucial. Interestingly, APP has also been implicated in brain metastasis despite having primary roles in the pathologies of neurodegenerative disease such as AD [69]. In vivo application of 75 mg/kg/day of LY-288671 in mice significantly reduced Aβ protein level and brain metastatic burden in both patient-derived cell cultures and an established melanoma cell line [69]. Novel connection between AD and brain melanoma metastasis suggests the emergence of future multi-targeting therapeutics for APP-associated pathologies. Although more research and clinical testing is needed regarding this topic, there is great promise for clinical application of LY-288671 and other similar β-secretase inhibitors in both AD and metastatic brain cancer mechanisms.

## 4. Gamma-Secretase Inhibitors

In line with targeting causes of the pathological accumulation of Aβ plaque in AD progression, the protease γ-secretase serves as another spotlight for therapeutic research within the field. This protein has four subunits that piece together to function in cleaving the transmembrane domains of over 100 membrane proteins [70]. In the amyloidogenic pathway of the amyloid cascade hypothesis, it was thought that γ-secretase was the final step before Aβ plaque formation, cleaving intramembrane segments on the fragment created by the β-secretase enzyme (also see Figure 1) [71]. Attempts at exploiting the role of γ-secretase in AD were trialed under the therapeutic efforts of γ-secretase inhibitors. Subsequently, the discoveries and contributions of this drug class will serve as the focus for this section.

The four subunits composing γ-secretase include presenilin (Psn), nicastrin (Nct), anterior- pharynx-defective-1 (Aph-1), and presenilin enhancer-2 (Pen-2) [72]. In the broad scheme of AD, presenilin proteins are of particular pathophysiologic importance due to dominant mutations that cause increased accumulation of Aβ42 and Aβ40 [73], consequently contributing to early-onset familial AD [74]. With regard to γ-secretase, the presenilin protein plays an integral role in enzymatic function as it is the catalyst for the function of this protease complex [71]. As a multi-transmembrane protein in itself, Psn forms its own molecular mass complex that is easily degraded [75]. However, when this holoprotein Psn is stabilized by Aph-1 and Nct, it forms a stable high molecular mass complex that prevents its degradation yet still renders the complex enzymatically inactive. Only when Pen-2 lyses the high molecular mass complex and presents the active aspartate residues of Psn does the structure come together as the active γ-secretase enzyme [75]. It is important to note that due to the multiplex nature of the γ-secretase complex, this protein is able to perform an array of different functions in unalike cells [76].

γ-secretase is unspecific and has many lipid-based intramembrane substrates that it targets, particularly among type I transmembrane proteins. APP was the first substrate demonstrated to be targeted by γ-secretase [76]. γ-secretase targets the c-terminal of the Aβ domain of APP after upstream processing by α- and β-secretases, releasing Aβ and APP intracellular domain (AICD) [27]. While the role of AICD in signal transduction is controversial [77], there is evidence that AICD is produced by the action of γ-secretase at the plasma membrane/early endosome location of cells [78]. Thus, while scientist is still unsure whether this plays a role in active disease pathophysiology or whether it is a downstream effect of AD, knowing where γ-secretase works helps to localize exactly where enzyme action occurs in cells and as a result, helps to determine where amyloid is formed.

Another notable target of γ-secretase includes transcription factor cyclic adenosine monophosphate response element binding (CREB) binding protein (CBP) [79]. This protein has an important modulatory role as a transcription repressor. Under mutations that disallow transcription repression, unregulated proliferation is then permitted to ensue, contributing to disease progression. γ-secretase also has an important relationship with the Notch receptor [80]. Notch is a ligand receptor that transduces signals allowing for cell differentiation; through proteolysis by γ-secretase, the notch intracellular domain (NICD) is produced and signals for neuron development and differentiation [80]. Along similar lines, ErbB-4 is a tyrosine kinase receptor that allows for cell proliferation. It has been demonstrated that γ-secretase cleaves the intracellular domain of this protein, similar to its mechanism with Notch, allowing for nuclear translocation and downstream signaling [81]. This is similar to the action described above with AICD, though the role of AICD in cell differentiation is still uncertain [78].

The most well-known drug among γ-secretase inhibitors is semagacestat (LY-450139). The main mechanism behind semagacestat is its role in inhibiting APP cleavage, with a dose-dependent effect of decreased Aβ plaque accumulation in the CNS [82]. Another study also found that semagacestat has an effect on reducing the density of dendritic spines in WT mice through modulation of APP cleavage [83]. While these findings were promising, they did not provide an indication for determining clinically significant γ-secretase inhibition in reducing the burden of disease [84]. By consequence of pathway signaling, the decreased Aβ accumulation created by semagacestat causes a countering rise in amyloid precursor protein c-terminal fragments (APP-CTF). Furthermore, due to the nonspecificity of γ-secretase, semagacestat decreased Notch signaling among other tissue types [82]. In clinical trials, this presented in participants as side effects including increased risk of skin cancer, gastrointestinal bleeding, and counterproductive worsened cognitive function [82]. Resultantly, Phase III clinical trials were paused on semagacestat.

Due to the side effects seen from inhibition of Notch signaling among γ-secretase inhibitors, another group in this drug class was developed; the new subtype was termed “Notch-sparing.” Specifically, avagacestat (BMS-708163) and begacestat (γ-secretase inhibitors-953) were two of the important names for this field. Though avagacestat showed decreased Aβ levels in the CSF of healthy participants, it did not show as much selectivity for APP over Notch as hoped. This was due to the fact that it bound the presenilin-1 N-terminal fragment and was able to block four different active-site probes [85]. In addition, there was not enough data to determine whether clinically significant reductions in mice brain plaque deposition occurred, as well as behavioral effects in animal-replicated AD [71]. Avagacestat was stopped in Phase II of clinical trials. Begacestat showcased different findings from avagacestat. Though classified under the same umbrella of Notch-sparing, the primary effect of begacestat was through improving memory in transgenic mouse models of AD [71]. Contrary to avagacestat, begacestat did not show a decrease in Aβ40 levels in the CSF of AD patients. Begacestat was also stopped in Phase II of clinical trials [71]. Other Notch-sparing γ-secretase inhibitors that were trialed in early stages include E2012 and PF-308414. E2012 was halted in Phase I trials due to its toxic side effect of lenticular opacity in rats, whereas PF-308414 was abandoned despite promising characteristics including good brain penetration and lack of rebound plasma Aβ [71]. Despite its disregard in AD therapeutics, PF-308414, among other γ-secretase inhibitors, is being trialed as a cancer therapeutic due to its inhibitory effect on Notch signaling [86].

While the premise of γ-secretase inhibitors was exciting for the field of AD therapeutics due to effects on decreasing Aβ levels, lack of clinical significance and non-selectivity of γ-secretase were the primary downfalls for this drug class. Furthermore, low doses of γ-secretase inhibitors showed a paradoxical increase in Aβ accumulation, and discontinuation was associated with rebound Aβ levels [87]. Forward research on the relationship between γ-secretase modulator proteins (GSMPs) and γ-secretase modulators (GSMs) with γ-secretase inhibitors could provide direction on the potential benefit of this drug in AD therapeutics. In addition, a better understanding of the effect of γ-secretase inhibitors on different substrates and in different tissues would promote recognition of its full function and potential side effects [87].

## 5. Amylin Agonists

Faced with data that challenges the AH, researchers have reassessed the function of amylin, also known as islet amyloid polypeptide, in the pathology of AD. Human amylin was first found in the pancreas, where it is co-secreted with insulin from pancreatic B cells [88]. Endocrinology research discovered that in the early stages of Type II diabetes mellitus, amylin levels are higher than usual. Additionally, this protein is prone to misfold and then form oligomers and fibrils when it is without a matched amount of stabilizing insulin [89,90]. Taking things a step further, data have shown that type 2 diabetes mellitus is a major risk factor for the development of AD [91]. Research has actually shown that these peptides cause death of neurons via induction of proapoptotic genes in a mechanistically similar way as Aβ plaques [92]. Outside of pathology, amylin reduces food intake and body weight, in addition to modulating nociception and cognitive function [92].

These findings paint the picture that amylin, as with Aβ peptides, are cytotoxic to neurons and pathogenic leading to AD. However, some findings suggest otherwise. For example, pramlintide—a synthetic amylin analogue—has been reported to attenuate both Aβ and amylin induced depression of LTP in the hippocampus of AD mice [93]. A noted amylin receptor antagonist (AC253) produces the same attenuation [92]. This apparent contrast puzzles researchers and has led to at least one group initially hypothesizing that there may be ‘biased agonism’, as has been recently reported for the calcitonin receptor (CTR) component of the amylin receptor (AMY). Regardless of the mechanism in question, follow-up experiments involving the administration of amylin and pramlintide in transgenic mouse models of AD lead to improvement in behavioral measures and an efflux of brain Aβ [94]. It is theorized that the excess amylin may act as a ‘peripheral sink’ which leads to the exodus of amyloid across the BBB in addition to amylin receptor interaction [92].

More investigation into pramlintide shows that the molecule increases the expression of proteins associated with synaptic plasticity and cognition [95]. Another study demonstrated a clear neuroprotective effect from pramlintide in intracerebroventricular injection of streptozotocin (to model sporadic AD) injected rat model of AD [96]. Interestingly, metformin was found to produce the same benefit in various experimental outcomes with the exception of recognition memory tests where it was inferior to pramlintide. This suggests that amylin might actually serve as a symptom alleviator for AD and lead to improved cognition [96].

Even more recent findings showed that oral administration of amylin, in addition to already established findings of intraperitoneal injections, reduce AD pathology at the cellular and behavioral level [97]. For hypothetical mechanisms of action, human amylin has been shown to modulate proteins and modify the expression of genes like cFos that are involved in synaptic plasticity [98,99,100]. In total, the evidence suggests that amylin is capable of changing signaling cascades and markers of synaptic plasticity, firmly placing this protein in the physiology of healthy cognition [93,98]. Not only that, but investigation has demonstrated that amylin significantly reduced CDK5 signaling which led to a reduction in tau phosphorylation, providing evidence that it may be beneficial in the treatment of AD via a variety of mechanisms [101]. At the very least, these data show that the amyloid hypothesis in its simplest form: fewer Aβ aggregates is good and more Aβ aggregates is bad. Thus, the beneficial effects of amylin addition will not violate the amyloid hypothesis. At the most, these data support the claim that the field’s resources should be poured into further delineating the varied responses of amylin receptors throughout the body, and the agonists that have varied stimulating effects on them in the pathology of AD and other protein folding diseases—such as Parkinson’s disorder or diabetes [102].

## 6. Discussion

The prevalence of AD continues to grow globally, and this is further compounded by its physical, emotional, and economic consequences on patients and their communities [103]. These consequences further substantiate the current imperative for therapeutic innovation [21,104,105,106,107,108,109]. The aim of this review was to examine a particular genre of AD therapeutics in the form of amylin small peptide modulation (see Table 1).

**Table 1 biomolecules-12-00996-t001:** Summary of therapeutic modulations of secretase and amylin for AD.

Biomolecule	Therapeutic Mechanism	Synthetic Subtypes under Investigation	Status of Investigations
α-secretase	Activation	ADAM10 ADAM17 Gemfibrozil	Gemfibrozil Phase I Trial—2019 [110,111] Acitretin Phase II Trial—2018 [112] APH-1105 Phase II Trial—2021 [113] Epigallocatechin-Gallate Phase III—2021 [114]
β-secretase	Inhibition	BACE1 LY-2886721	AZD3293 Phase I Trial—2014 [115] LY-2886721 Phase II Trial—2018 [116] JNJ-54861911 Phase II Trial—2022 [117,118] CNP520 Phase II Trial [119] Verubecestat Phase III Trial—2019 [120]
γ-secretase	Inhibition	LY-450139 E2012 PF-308414	LY-450139 Phase III Trial—2019 [121] Semagacestat Phase III Trial—2014 [122] Avagacestat Phase II Trial—2015 [123] GSI-136 Phase I Trial—2010 [124] NGP-555 Phase I Trial—2016 [125]
Amylin	Agonist	Pramlintide acetate Exenatide	Exendin-4 Phase II Trial—2018 [126]

The intrigue to critically review small peptides is based on the enormous versatility which these biomolecules have to maintain structural stability while penetrating membrane layers to induce an array of therapeutic effects [127,128]. This is in addition to the lower production costs of generating these therapies [129,130]. Specifically, amylin-based small peptides provides the greatest breadth of literature on investigation and was therefore the scope of this review [131,132]. However, there are other small peptide therapies which have been investigated as amylin modulators which were not discussed, such as humanin. This was due to the paucity of contemporary literature to support these peptides compared to the modulators discussed in this review [133,134,135,136,137]. In addition, while this review was built of the foundation created by the amyloid cascade hypothesis, there are other theories investigated in the literature on the pathophysiology of AD as well [138,139,140,141]. In particular, the tau hypothesis of AD is worth mentioning. As with amyloid, dysfunctional accumulation (e.g., tau) leads to effects in neuronal transmission [131,142,143,144,145,146,147]. This hypothesis is much newer, but there are small peptide modulation investigations occurring related to this hypothesis as well [143,148,149].

Table 1 outlines the current status of clinical trials among these agents by targeting secretase and amylin [67,110,120,121,122,123,126,150]. Moreover, the findings of this review with regard to α-secretase activators, β-secretase inhibitors, γ-secretase inhibitors, and amylin agonists align with the findings of previous reviews but also build upon with recent evidence on small peptide therapeutics as well [27,72,149,151,152,153,154,155,156,157,158,159,160,161,162,163,164,165,166,167]. This is encouraging given the current development of newer of AD therapeutics outside of small peptides. Of note, aducanumab is one of the most recently approved agents by Food and Drug Administration (FDA) for the management of AD. This agent functions as a monoclonal antibody which targets amylin-beta plaques in the brain. Its preclinical studies revealed decreased levels of chimeric Aβ, and a greater binding affinity to the fibrillary form rather than monomeric form. Moreover, this became the basis of targeting the Aβ precursors itself, rather than when the Aβ plaques started to form [168,169]. Further evidence and post-market monitoring is still in development after results from the EMERGE and ENGAGE phase 3 clinical trials led to FDA approval [170,171,172,173,174]. However, further exploration and post hoc analysis on the outcomes of these results suggested futility compared to initial reviews [175,176]. The timeline of these observations even led to adjustments on the approval’s intended audience where initial approval was for anyone with AD, but now it is for mild cognitive impairment or mild dementia stage [177,178,179,180,181,182,183,184,185,186,187,188]. This further opens the door for additional modulators such as small peptides to continue investigation and therapeutic development. As outlined in Section 2, the role of retinoids has primarily been investigated through its relationship with α-secretase. However, there is still evidence of retinoid involvement (e.g., all-trans-retinoic-acid) which activates protein kinase C and subsequently regulates β-secretase trafficking [189,190,191,192,193,194]. Similarly, retinoic acid may also have a role in γ-secretase activity [195]. This activity may be primarily affected during the synthesis of γ-secretase, which this process has been known to be inhibited by extracellular signal-regulated kinases (ERKs) [196] via activation of ERKs and phosphorylation through the mitogen-activated protein kinase cascade [197]. Let alone, retinoic acid has been experimentally shown to increases ERK phosphorylation in a dose-dependent manner [195].

As outlined in Section 3, investigation on LY288671 BACE1 inhibition has been minimal in comparison to other ongoing β-secretase inhibitor (i.e., verubecestat) trials [198,199]. Moreover, in vitro expression of these agents has also been found to be decrease in exposure to tocopherol derivatives [200]. Although the current body of literature is limited to transcriptional studies [200,201,202,203,204,205,206,207], but does gain support from historical animal models with a broader scope on Vitamin E treatment for AD [193,194,195,196,197,203,204,205,206,207]. This evidence further justifies the need for greater clinical trial coverage of vitamin E treatment. As outlined in Section 4, the evidence of paradoxical increase in Aβ accumulation via low dose of γ-secretase inhibitors and discontinuation [87] has made this category of small peptide modulators less feasible in comparison. Regardless, there remains a need for further trial investigation with tighter control variables as the previous trials may have lacked this to a degree given their discontinued status [208,209,210,211]. As outlined in Section 5, the role of pramlintide for the treatment of AD is promising in concept. However, there is a lack of randomized clinical trials on this purpose to date. Regardless, the cost effectiveness of this highly used diabetic medication creates great implementation potential for this agent [88,212,213,214,215].

In conclusion, the past decade has experienced tremendous growth in the degree of investigation for AD therapeutics. This growth has also led to development of small peptide therapeutics designed to modulate the pathophysiology of disease process. Of the many investigations conducted in the past decade, amylin agonists and modulators of α-, β-, and γ-secretases seem to hold the great promise based on their tremendous body of literature synthesized in this review. Future clinical trials can provide the translational evidence to further reveal the promised behind these drug developments.

## Figures and Tables

**Figure 1 biomolecules-12-00996-f001:**
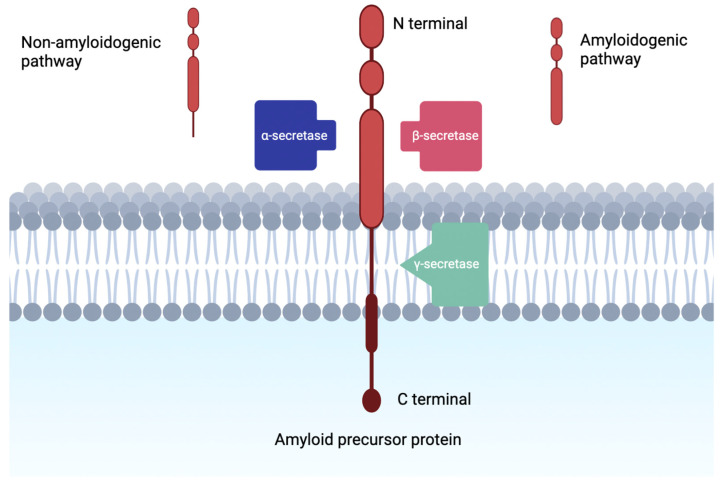
Visualization of amyloidogenic and non-amyloidogenic pathways in the cleavage of amyloid precursor protein. Notice the intramembranous piecing produced by the non-amyloidogenic pathway.

## Data Availability

Not applicable.

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
