# Peer review of "Amylin and Secretases in the Pathology and Treatment of Alzheimer’s Disease"

_biomolecules, 2022, doi:10.3390/biom12070996_

Round 1

Reviewer 1 Report

The manuscript entitled “Therapeutic Potential of Amylin Small-Peptide Modulation in Alzheimer’s Disease” describes one of the hypotheses in the pathophysiology of Alzheimer's disease, the beta- amyloid cascade hypothesis.

Generally speaking, according to this hypothesis, amyloid plaque accumulation is associated with disease progression.

Small peptides such as the three secretases (alpha, beta, gamma) and amylin are involved in many physiological processes and their disturbance can lead to neuronal damage and, consequently, to dementia.

The authors detail these processes and possible therapeutic intervention through the use of compounds that act as alpha-secretase activators, beta-secretase inhibitors, gamma-secretase inhibitors or amylin agonists.

This review work is interesting and valuable.

I propose a manuscript correction.

The title only emphasizes the role of amylin and omits the widely described secretases.

Table 1 should be completed with more detailed information about clinical trials - for example:

Alpha secretase activator gemfibrozil (chemical structure), current status of clinical trials (Phase II, continued or discontinued, data, year)

Beta-secretase inhibitors – verubecestat, LY288671 – look at the comment above  

Gamma-secretase inhibitors – semagacestat, avagacestat, begacestat - look at the comment above

I also suggest adding a little more information on aducanumab.

Author Response

The manuscript entitled “Therapeutic Potential of Amylin Small-Peptide Modulation in Alzheimer’s Disease” describes one of the hypotheses in the pathophysiology of Alzheimer's disease, the beta- amyloid cascade hypothesis.

Generally speaking, according to this hypothesis, amyloid plaque accumulation is associated with disease progression.

Small peptides such as the three secretases (alpha, beta, gamma) and amylin are involved in many physiological processes and their disturbance can lead to neuronal damage and, consequently, to dementia.

The authors detail these processes and possible therapeutic intervention through the use of compounds that act as alpha-secretase activators, beta-secretase inhibitors, gamma-secretase inhibitors or amylin agonists.

This review work is interesting and valuable.

I propose a manuscript correction.

The title only emphasizes the role of amylin and omits the widely described secretases.

Response: Thank you for your great comments and suggestions! The title has been changed into “Amylin and Secretases in the Pathology and Treatment of Alzheimer’s Disease”.

Table 1 should be completed with more detailed information about clinical trials - for example:

Alpha secretase activator gemfibrozil (chemical structure), current status of clinical trials (Phase II, continued or discontinued, data, year)

Beta-secretase inhibitors – verubecestat, LY288671 – look at the comment above 

Gamma-secretase inhibitors – semagacestat, avagacestat, begacestat - look at the comment above

Response: We appreciate your suggestion and have revised Table 1 as desired. The revised Table 1 includes the noteworthy clinical trials which were performed in the United States. Furthermore, we have added references and point out these trials in our discussion section.

I also suggest adding a little more information on aducanumab.

Response: Thank you for your great point. We have added more discussion and citations between 413-424. You also can see it below:

Of note, aducanumab is one of the most recently approved agents by Food and Drug Administration (FDA) for the management of AD. This agent functions as a monoclonal antibody which targets amylin-beta plaques in the brain. Its preclinical studies revealed decreased levels of chimeric Aβ, and a greater binding affinity to the fibrillary form rather than monomeric form. Moreover, this became the basis of targeting the Aβ precursors itself, rather than when the Aβ plaques started to form [158,159]. Further evidence and post-market monitoring is still in development after results from the EMERGE and ENGAGE phase 3 clinical trials led to FDA approval [160–164]. However, further exploration and post hoc analysis on the outcomes of these results suggested futility compared to initial reviews [165,166]. The timeline of these observations even led to adjustments on the approval’s intended audience where initial approval was for anyone with AD, but now it is for mild cognitive impairment or mild dementia stage [167–178].

Reviewer 2 Report

The manuscript title is misleading. The authors mainly described APP processing but not the therapeutic potential of amylin. They should change the title.

In line 15-16, they described “Among these small peptides, there are 4 modulatory mechanisms of interest in this review: alpha-, beta-, gamma-secretases and amylin.” However, alpha-, beta-, gamma-secretases are proteases but not peptides.

In line 156, Abeta plague should be Abeta plaque.

In line 162, they described that “A673T mutations of BACE1”. However, A673T is a polymorphism of APP but not BACE1.

In line 161, they described “One such study involved a naturally occurring, partial BACE1 KO seen in Icelandic populations [47].” However, in the 47 reference, the A673T Icelandic polymorphisms but not BACE1 KO is described. Thus the reference should be wrong.

There are many other flows in the manuscript. They should re-visit the manuscript and carefully check at multiple levels, spelling, misleading, wrong references, and the logics.

Author Response

The manuscript title is misleading. The authors mainly described APP processing but not the therapeutic potential of amylin. They should change the title.

Response: Thank you for your great point. The title has been changed into “Amylin and Secretases in the Pathology and Treatment of Alzheimer’s Disease”.

In line 15-16, they described “Among these small peptides, there are 4 modulatory mechanisms of interest in this review: alpha-, beta-, gamma-secretases and amylin.” However, alpha-, beta-, gamma-secretases are proteases but not peptides.

Response: Great point and thank you. We have adjusted terminology use in this paragraph to explain that these secretases are not peptides. Moreover, we have utilized the term biomolecule, where appropriate, in order to reduce ambiguity between peptide/small peptide/protein.

In line 156, Abeta plague should be Abeta plaque.

Response: Changed as suggested.

In line 162, they described that “A673T mutations of BACE1”. However, A673T is a polymorphism of APP but not BACE1.

Response:  Great point. This has been adjusted to APP. We have revised this section in order to provide greater clarity to the readers. Please see line 163 and line 164.

In line 161, they described “One such study involved a naturally occurring, partial BACE1 KO seen in Icelandic populations [47].” However, in the 47 reference, the A673T Icelandic polymorphisms but not BACE1 KO is described. Thus the reference should be wrong.

Response: Thank you for this key input. This has been changed. The reference should be cited from Jonsson et al. 2012 (please see reference 48).

There are many other flows in the manuscript. They should re-visit the manuscript and carefully check at multiple levels, spelling, misleading, wrong references, and the logics.

Response: We appreciate your critical comments and made it a priority to revisit the manuscript for the sake of clarity. This process began by reviewing the manuscript with 3 native English speakers. Thereafter, references were adjusted where necessary and new references were added in areas to improve the perspective which our manuscript tries to make. The total reference count is now 205. Moreover, there was greater revision place on discussion of aducanumab to improve its claim and clarity. Original figure 1 was removed to limit any unnecessary additions/improve clarity. New figure 1 was provided more detailed information for clinical trials. Table 1 was revised a great deal to add status of noteworthy clinical trials, agents of interest, and references were added in discussion section. Thank you!

Round 2

Reviewer 2 Report

The authors extensively revised the manuscript and improve the quality. However, I still have some concerns to be addressed before publication.

The authors still describe “One such study involved a naturally occurring, partial BACE1 gene KO seen in Icelandic populations [48]. ” However, in the reference 48, Icelandic mutation, A673T in APP but not BACE1, was described. Please refer suitable references.

The authors describe “At the very least, these data show that the amyloid hypothesis in its simplest form - less amyloid good more amyloid bad - is blatantly wrong.” However, in amyloid hypothesis, only amyloid beta but not other amyloid is considered. Thus, “less amyloid good more amyloid bad” is not claimed in the amyloid hypothesis. What amyloid hypothesis claim is that less amyloid-beta aggregates good more amyloid-beta aggregates bad. Thus, the beneficial effects of amylin addition will not violate the amyloid hypothesis.

Author Response

The authors extensively revised the manuscript and improve the quality. However, I still have some concerns to be addressed before publication.

The authors still describe “One such study involved a naturally occurring, partial BACE1 gene KO seen in Icelandic populations [48]. ” However, in the reference 48, Icelandic mutation, A673T in APP but not BACE1, was described. Please refer suitable references.

Response: Thanks for your great point. We deleted this reference and cited right one for our newly description (please see lines 162 to 164). Please also see it below:

One such study demonstrated that partial BACE1 gene deletion (50% to 70%) induces synaptic plasticity deficit in adult mice [48].

Reference 48: Lombardo, S.; Chiacchiaretta, M.; Tarr, A.; Kim, W.; Cao, T.; Sigal, G.; Rosahl, T.W.; Xia, W.; Haydon, P.G.; Kennedy, M.E.; et al. BACE1 Partial Deletion Induces Synaptic Plasticity Deficit in Adult Mice. Sci. Rep. 2019; 9(1):19877. doi: 10.1038/s41598-019-56329-7.

The authors describe “At the very least, these data show that the amyloid hypothesis in its simplest form - less amyloid good more amyloid bad - is blatantly wrong.” However, in amyloid hypothesis, only amyloid beta but not other amyloid is considered. Thus, “less amyloid good more amyloid bad” is not claimed in the amyloid hypothesis. What amyloid hypothesis claim is that less amyloid-beta aggregates good more amyloid-beta aggregates bad. Thus, the beneficial effects of amylin addition will not violate the amyloid hypothesis.

Response: Great point. We revised the statement as suggested (see lines 377 to 380). Please also see it below:

At the very least, these data show that the amyloid hypothesis in its simplest form: less Aβ aggregates good more Aβ aggregates bad. Thus, the beneficial effects of amylin addition will not violate the amyloid hypothesis.